

# The vaginal metabolomics profile with features of polycystic ovary syndrome: a pilot investigation in China

Yan Xuan[1], Xiang Hong[1], Xu Zhou[1], Tao Yan[1], Pengfei Qin[2], Danhong Peng[3] and Bei Wang[1]

[1] Department of Epidemiology and Health Statistics, Southeast University, Nanjing, Jiangsu, China
[2] Nanjing Women and Children's Healthcare Hospital, The Affiliated Obstetrics and Gynecology Hospital with Nanjing Medical University, Nanjing, Jiangsu, China
[3] Department of Obstetrics and Gynecology, Zhongda Hospital, School of Medicine, Southeast University, Nanjing, China

Corresponding author
Bei Wang, wangbeilxb@163.com

## ABSTRACT

**Background**. Polycystic ovary syndrome (PCOS) is the most common metabolic disorder and reproductive endocrine disease, posing an elevated risk to women of reproductive age. Although metabolism differences in serum, amniotic fluid and urine have been documented in PCOS, there remains a paucity of evidence for vaginal fluid. This study aimed to identify the metabolic characteristics and potential biomarkers of PCOS in Chinese women of reproductive age.

**Methods**. We involved ten newly diagnosed PCOS women who attended gynecology at Zhongda Hospital and matched them with ten healthy controls who conducted health check-up programs at Gulou Maternal and Child Health Center in Nanjing, China from January 1st, 2019 to July 31st, 2020. Non-targeted metabolomics based on ultra-high-performance liquid chromatography tandem mass spectrometry (UHPLC-MS/MS) was applied to differentially screen vaginal metabolites between PCOS group and healthy controls. Principal component analysis (PCA), orthogonal partial least-squares discriminant analysis (OPLS-DA) and enrichment analysis were used to observe differences, search for potential biomarkers and enrich related pathways.

**Results**. Among the 20 participants, a total of 195 different metabolites were detected between PCOS group and healthy control group. PCOS and control groups were effectively separated by vaginal fluid. Lipids and lipid-like molecules constituted the majority of differential metabolites. Notably, dopamine exhibited an increased trend in PCOS group and emerged as the most significant differential metabolite, suggesting its potential as a biomarker for identifying PCOS. The application of UHPLC-MS/MS based vaginal metabolomics methods showed significant differences between PCOS and non-PCOS healthy control groups, especially linoleic acid metabolism disorder. Most differential metabolites were enriched in pathways associated with linoleic acid metabolism, phenylalanine metabolism, tyrosine metabolism, nicotinate and nicotinamide metabolism or arachidonic acid metabolism.

**Conclusions**. In this pilot investigation, significant metabolomics differences could be obtained between PCOS and healthy control groups. For PCOS women of reproductive age, vaginal metabolism is a more economical, convenient and harmless alternative to provide careful personalized health diagnosis and potential targets for therapeutic intervention.

# INTRODUCTION

Polycystic ovary syndrome (PCOS) is a most common heterogeneous, female endocrine-reproductive-metabolic abnormality (*Sun et al., 2021*). It has significant clinical features of hyperandrogenism, ovulation dysfunction and polycystic morphology of ovaries (*Rotterdam ESHRE/ASRM-Sponsored PCOS consensus workshop group, 2004*) and affects over 200 million reproductive age women globally (*Parker et al., 2022*). The incidence rate of PCOS is the highest in Ecuador, and has the steepest increases in Brazil, Ethiopia and China, leading it to a global public health concern (*Liu et al., 2021*). Moreover, PCOS is related to insulin resistance (*Moghetti & Tosi, 2021*), obesity (*Simkova et al., 2020*), type 2 diabetes mellitus (*Livadas et al., 2022*), cardiovascular diseases (*Livadas et al., 2022*) and is one cause of infertility (*Pericuesta et al., 2020*) in reproductive age women. Compelling evidence (*Basirat et al., 2019*; *Dovom et al., 2023*; *Subramanian et al., 2022*; *Wu et al., 2020*; *Xing et al., 2022*) has suggested that women with PCOS are vulnerable to hypertension, nephrolithiasis, depression and pregnancy complications such as miscarriage and preterm birth, accompanied by 62% (*Wu et al., 2020*), 59% (*Dovom et al., 2023*), 57% (*Xing et al., 2022*), 33% (*Basirat et al., 2019*) and 13% (*Subramanian et al., 2022*) increase odds among women with PCOS, respectively. Due to the adverse effects on female fertility, most studies mainly focus on women of reproductive age, especially among those between 15 and 49 years old (*World Health Organization, 0000*). Overall, PCOS is closely related to women's physical, mental health and future pregnancy outcomes.

Currently, the diagnoses of PCOS (*Medical AOaGBotC, 2018*) are still not uniform across countries because of the unclear etiology of PCOS, which fairly does not conduce to a prediction and prevention of the disease. In practice, PCOS is a diagnosis of exclusion according to the disease guidelines (*Shabbir et al., 2023*). Although PCOS is commonly seen clinically, its underlying mechanism and pathogenesis have not been fully elucidated. Hypotheses are mainly focused on epigenetic factors such as non-coding RNAs and DNA methylation which may be involved with dopaminergic synapse (*Cao et al., 2021*; *Chen et al., 2021*; *Mu et al., 2021*), the susceptibility variants from genome-wide association studies by next-generation sequencing (*Dapas & Dunaif, 2022*), environmental exposure (*Vilarino-Garcia et al., 2022*) and the imbalance of vaginal microbiota (*Hong et al., 2020*). Therefore, providing another understandable and reliable perspective to clarify the molecular mechanism of PCOS has become both a clinical and a scientific challenge. Metabolomics has become a research tool for accessing changes in organisms (*AbuBakar Sajak et al., 2021*) and has been used in the etiological study of PCOS and the search for biomarkers (*Koivula et al., 2019*). Previous studies have been conducted in human serum (*Ozegowska et al., 2021*), follicular fluid (*Chen et al., 2020*) or urine samples in PCOS patients (*Dhayat et al., 2018*) and focus on disturbed metabolism of the amino acids, carbohydrates, lipids, *etc.* However, there are inevitable limitations. Serum samples

mainly focus on lipid mentalism and the withdrawal practice is invasive, follicular fluid is usually used in women undergoing *in vitro* fertilization treatment and the withdrawal is also inevitably invasive, and urine samples usually own a limited number of metabolites. Vaginal discharge metabolomics is a widely used approach to identify cervical cancer (*Xu et al., 2022*), vaginal infection (*Borgogna et al., 2020*) and placental abruption (*Gelaye et al., 2016*) due to its potential capacity of being the top predictor of cervical vaginal environment status and female reproductive system (*Bokulich et al., 2022*). Meanwhile, alteration in dopamine turnover may be related to endocrine disorders (*Ianosi et al., 2016*), which is closely correlated with PCOS and can be directly detected in vaginal secretions. Vaginal secretion metabolomics among PCOS patients, which is a more economical, convenient and harmless method (*Lu et al., 2021*), has been reported minimally (*Ozegowska et al., 2021*). To update current knowledge on different biological samples of PCOS, we focused on Chinese PCOS female's vaginal secretion to identify correlated metabolic factors. As a supplementary analytical technique of nuclear magnetic resonance (NMR) and gas chromatography-mass spectrometry (GC-MS), most ultra-high-performance liquid chromatography tandem mass spectrometry (UHPLC-MS/MS) non-target metabolomics technology has high sensitivity, good retention time reproducibility and extensive chemical diversity coverage.

In this study, we conducted a pilot investigation among PCOS women between 20 and 45 years old and used UHPLC-MS/MS to analyze metabolic profiles of reproductive secretions. We aimed to reveal the correlation between clinical features and vaginal metabolites and to identify the metabolic characteristics and potential biomarkers of PCOS in Chinese women of reproductive age, which may help us comprehensively understand the molecular mechanism and easily diagnose PCOS.

## MATERIALS AND METHODS

### Study participants and procedure

This pilot investigation was reviewed and initiated by Zhongda Hospital and the Gulou Maternal and Child Health Center in Nanjing, China to improve women's health and fecundability. We initially recruited 287 women at gynecology in Zhongda Hospital and 523 health check-up women which was similar to the general population in Gulou Maternal and Child Health Center from January 1, 2019 to July 31, 2020. After signing written informed consent, they completed health checkups and examinations, including body mass index (BMI), fasting plasma glucose, menstruation period, dysmenorrhea, vaginal cleanness, vaginal pH and bacterial vaginosis infection.

### Inclusion criteria

We included PCOS women who met the following criteria: (1) aged 20–45 years; (2) were willing to be followed until a confirmed diagnosis of PCOS, which followed both the 2004 revised Rotterdam criteria (*Rotterdam ESHRE/ASRM-Sponsored PCOS consensus workshop group, 2004*) and the Guidelines for Diagnosis of PCOS in Chinese (2018 Edition) (*Endocrinology Subgroup and Expert Panel CSoOaG et al. , 2018*). During the contemporaneous period, women recruited for the healthy control group were 1:1

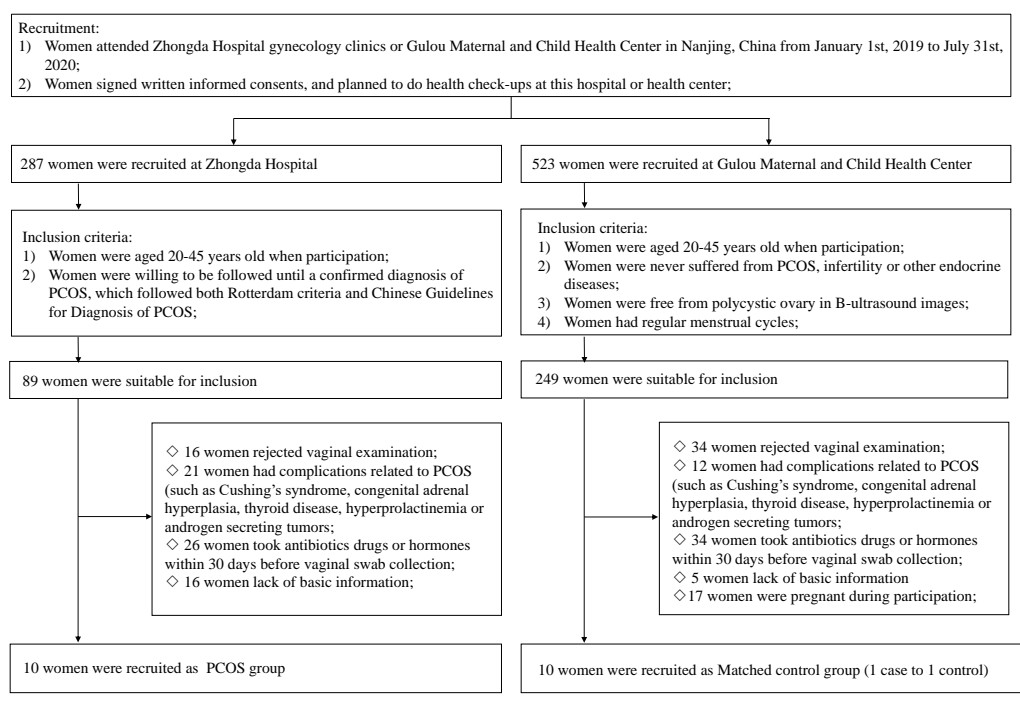

**Figure 1** **The flowchart for the study.** Abbreviations: PCOS, polycystic ovary syndrome.

matched by age (±3 years old) and ethnicity, owned the inclusion criteria as follows: (1) the ages of women were between 20 and 45; (2) had never suffered from PCOS, infertility or other endocrine diseases; (3) no polycystic ovary was found in B-ultrasound images; (4) had regular menstrual cycles.

*Exclusion criteria*

Women of any group would be excluded if they (1) rejected vaginal examination when participation; (2) had following complications: Cushing's syndrome, congenital adrenal hyperplasia, thyroid disease, hyperprolactinemia or androgen secreting tumors; (3) took antibiotics, drugs or hormones within 30 days before vaginal examination; (4) lacked basic information; (5) were pregnant during participation (Fig. 1). Finally, ten women with PCOS and ten matched controls were included in this study. This study has been approved by the Ethics Committee of Zhongda Hospital (2018ZDSYLL072-P01), all patients have provided written informed consents.

All participants carried out basic information records the first time they reached the clinics or health center. After signing written informed consent, they completed health examinations and were recorded for menstrual intervals. For healthy controls, they were followed up every month for exact menstruation length. During the gap of menstruation period, health workers would confirm that women were not sexually active or had vaginal irrigation in the previous 48 h before vaginal swab collection. Swabs were collected from all participants at the lithotomy site from their posterior fornix by rotating three times. Collected swabs were stored in a drying tube with identification labels and then arranged

orderly into a special collection container at 4 °C. Furthermore, they were placed in a −80 °C refrigerator until metabolites detecting procedure was conducted.

## Basic information definition

Basic information was recorded based on previous studies on PCOS. For sociodemographic characteristics, age was classified into 20–24 years and ≥25 years. Occupation was categorized into worker, civil servant and others. Educational level was classified into bachelor degree and master degree or above. For health statuses, BMI was calculated by dividing weight by the square of height ($kg/m^2$), and was binary categorized ($\leq 24\,kg/m^2$ and $>24\,kg/m^2$) (*Ji & Chen, 2013*). Fasting plasma glucose was tested in the morning after 8 h of fasting (*Alberti & Zimmet, 1998*). Menstrual period for healthy controls was recorded for at least three menstruation cycles and averaged, and for women with PCOS, they self-reported their last menstruation period length. Menstruation period was divided into less than or equal to one month and more than one month. Self-reported dysmenorrhea was classified as yes or no. Vaginal cleanness (I °–II °, III °–IV °) and bacterial vaginosis infection (yes/no) were based on hospital laboratory testing.

## UHPLC-MS/MS, data processing and identification

All samples were thawed at 4 °C for extraction, which would be conducted with an additional one mL of extract solution (ACN:MeOH 50:50 (v/v), with isotopically-labelled internal standard mixture). The mixture was vortex-mixed for 30 s and then incubated in ice water for 30 min. The samples were homogenized at 35 Hz for 4 min, sonicated in ice water for 5 min, incubated at −40 °C for 1 h, and centrifuged at 12,000 rpm for 15 min at 4 °C. A 750 μL aliquot supernatant of each sample was prepared in EP tubes. The resulting supernatant was vaporized without heating, and a 100 -μL aliquot of extract solution (ACN: MeOH 50:50 (v/v)) was added to make up the dried sample. Next, the solution was vortexed for 30 s, sonicated in ice water for 10 min, and centrifuged at 4 °C for 15 min. The supernatant was moved into a fresh glass vial for UPLC-MS analysis. All samples were kept inside an auto-sampler with a temperature of 4 °C during the analysis.

The UHPLC system (Vanquish; Thermo Fisher Scientific) was used to conduct UHPLC-MS/MS analysis with a UPLC BEH Amide column (2.1 mm × 100 mm, 1.7 μm) combined to Q Exactive HFX mass spectrometer (Orbitrap MS; Thermo Fisher Scientific) with electrospray ionization (ESI). Quality control (QC) samples were used to monitor the reliability and stability. Peaks were normalized and filtered to remove noise based on relative standard deviation, and data with no more than 50% null values or no more than 50% in all groups were retained. The parameter cutoff was set to 0.5.

## Statistical analysis

We described baseline information in terms of means (standard deviation) and counts (percentages). Pairwise student's *t*-test and Fisher exact test were used to analyze data between groups. Pearson and Spearman correlation were used for correlation analysis. These analyses were conducted using R statistical software (version 4.1.3; *R Core Team, 2022*). For metabolomics analysis, principal component analysis (PCA) and orthogonal

partial least-squares discriminant analysis (OPLS-DA) were conducted using SIMCA software (version 16.0.2). A two-sided $p$ value <0.05 was considered statistically significant.

## RESULTS

In this study, ten PCOS and ten health control women were finally included. All of them were Han ethnicity as they accounted for the largest ethnicity group in China. After age-matching, the average ages of PCOS and health control women were $24.20 \pm 2.35$ and $25.20 \pm 3.39$ years, respectively. Compared to healthy controls, PCOS women presented more extended menstruation periods and higher vaginal pH (median 6.0, with first quartile of 5.63 and third quartile of 6.50) ($p < 0.05$). The sociodemographic characteristics and health status of the study population are presented in Table 1.

All samples were successfully measured by UHPLC-MS/MS. A total of 859 metabolites were detected in the study cohort. Among them, 195 differential metabolites were found between PCOS and healthy control, containing 122 metabolites in positive ion mode (75 compounds showed upward trends while 47 compounds showed downward trends) and 77 metabolites in negative ion mode (51 compounds showed upward trend while 26 compounds showed downward trend), in which four of them were detected both in ESI positive and negative modes. The differential compounds were categorized into seven major classes: alkaloids and derivatives, benzenoids, lipids and lipid-like molecules, organic acids and derivatives, organic oxygen compounds, organoheterocyclic compounds, phenylpropanoids and polyketides.

In unsupervised PCA score plots of the metabolite results (Fig. S1), an overall view of the two groups was initially revealed and all sample data were wrapped in the Hotelling $T$-squared ellipse. Meanwhile, both PCOS and control groups could be clearly separated, and QC samples were closely assembled to show the stability of the test results. Similarly, OPLS-DA score plots showed the same separation between the two groups with well robustness and no overfitting (Fig. 2).

For potential biomarker searching, the Euclidean distance matrix for the quantitative value of differential metabolite was calculated among all different compounds. Figure 3 shows the most significant 20 metabolites which changed remarkably. Among them, the majority were lipids and lipid-like molecules, followed by phenylpropanoids and polyketides, organic acids and derivatives, benzenoids and alkaloids and derivatives. Among the greatest differential metabolites, dopamine was the top significant metabolite in vaginal secretions with an upward trend in PCOS group, followed by increases of isocitric acid, 5,6-dimethoxysterigmatocystin, m-coumaric acid, oxoadipic acid, l-cysteine, 20-hydroxyeicosatetraenoic acid, 3,4-dihydroxymandelaldehyde, 3-o-feruloylquinic acid, 5,7-dihydroxy-2-phenyl-6,8-bis[3,4,5-trihydroxy-6-(hydroxymethyl)oxan-2-yl]-4h-chromen-4-one and sedoheptulose in PCOS group, while n-cyclopropyl-trans-2-cis-6-nonadienamide, 4,8 dimethylnonanoyl carnitine, 11-dehydro-thromboxane B2, 3,3-dimethylglutaric acid, 24-epibrassinolide, dioscoretine, 2-(3,4-dihydroxyphenyl)-3,4-dihydro-2h-1-benzopyran-3,5,7-triol, caprylic acid and 8-hydroxyoctanoate showed downward trends.
**Table 1  Baseline characteristics of the study population.**

| Variables | PCOS No. (%) | Control No. (%) | t | P |
|---|---|---|---|---|
| **All participants** | **10** | **10** | | |
| **Sociodemographic characteristics** | | | | |
| Age, years (mean ± SD) | 24.20 ± 2.35 | 25.20 ± 3.39 | 0.77 | 0.45 |
| 20–24 | 5 (50.00) | 5 (50.00) | – | 1.00 |
| ≥25 | 5 (50.00) | 5 (50.00) | | |
| Missing | 0 | 0 | | |
| Occupation | | | | |
| Worker | 8 (80.00) | 4 (40.00) | – | 0.15 |
| Civil servant | 2 (20.00) | 3 (30.00) | | |
| Others | 0 (0.00) | 3 (30.00) | | |
| Missing | 0 | 0 | | |
| Educational level | | | | |
| Bachelor degree | 8 (80.00) | 9 (90.00) | – | 1.00 |
| Master degree or above | 2 (20.00) | 1 (10.00) | | |
| Missing | 0 | 0 | | |
| **Health statuses** | | | | |
| BMI, kg/m$^2$ (mean ± SD) | 21.12 ± 4.46 | 22.01 ± 3.68 | −0.49 | 0.63 |
| Underweight/Normal (≤24) | 7 (70.00) | 8 (80.00) | – | 1.00 |
| Overweight/Obesity (>24) | 3 (30.00) | 2 (20.00) | | |
| Missing | 0 | 0 | | |
| Fasting plasma glucose, mmol/L (mean ± SD) | 4.78 ± 0.48 | 4.85 ± 0.49 | −0.32 | 0.75 |
| <6.1 | 10 (100.00) | 10 (100.00) | – | 1.00 |
| ≥6.1 | 0 (0.00) | 0 (0.00) | | |
| Missing | 0 | 0 | | |
| Menstruation period | | | | |
| ≤1 month | 0 (0.00) | 9 (90.00) | – | <0.001 |
| >1 month | 10 (100.00) | 1 (10.00) | | |
| Missing | 0 | 0 | | |
| Dysmenorrhea | | | | |
| Yes | 8 (80.00) | 5 (50.00) | – | 0.35 |
| No | 2 (20.00) | 5 (50.00) | | |
| Missing | 0 | 0 | | |
| Vaginal cleanness | | | | |
| I°–II° | 4 (40.00) | 8 (80.00) | – | 0.17 |
| III°–IV° | 6 (60.00) | 2 (20.00) | | |
| Missing | 0 | 0 | | |
| Vaginal pH (mean ± SD) | 6.10 ± 0.91 | 4.85 ± 0.36 | −4.05 | 0.002 |
| ≤5.5 | 3 (3.00) | 10 (100.00) | – | 0.003 |
| >5.5 | 7 (7.00) | 0 (0.00) | | |

**Table 1** (*continued*)

| Variables | PCOS<br>No. (%) | Control<br>No. (%) | t | P |
|---|---|---|---|---|
| Missing | 0 | 0 | | |
| Bacterial vaginosis infection | | | | |
| Yes | 2 (20.00) | 0 (0.00) | – | 0.47 |
| No | 8 (80.00) | 10 (100.00) | | |
| Missing | 0 | 0 | | |

**Notes.**

Data presented as *n*(%), unless noted otherwise.

Abbreviations: BMI, body mass index; SD, standard deviation; PCOS, polycystic ovary syndrome.

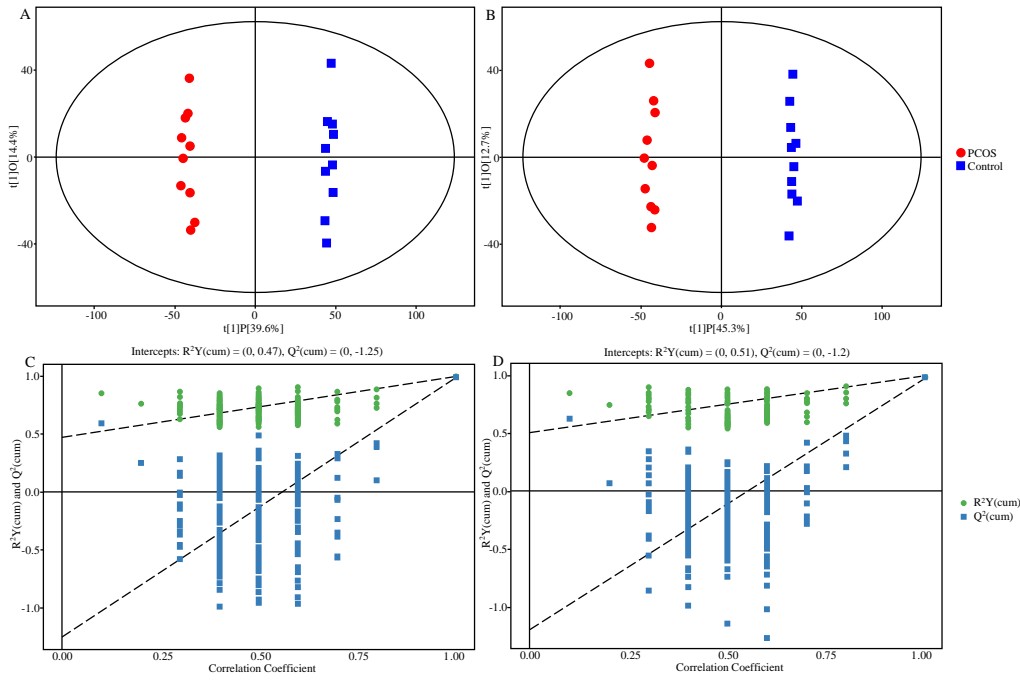

**Figure 2   Plots of OPLS-DA score and permutation test.** (A) OPLS-DA score plot in positive ion mode. (B) OPLS-DA score plot in negative ion mode. (C) OPLS-DA permutation test plot in positive ion mode, $R^2Y = 0.47$, $Q^2 = -1.25$. (D) OPLS-DA permutation test plot in negative ion mode, $R^2Y = 0.51$, $Q^2 = -1.20$. Abbreviations: OPLS-DA, orthogonal partial least-squares discriminant analysis; PCOS, polycystic ovary syndrome.

To expound on metabolic and regulatory pathways, all differential metabolites were searched in Kyoto Encyclopedia of Genes and Genomes (KEGG). The pathways were filtered based on bubble size and color in bubble plots (Fig. 4). The differential metabolites were enriched in five main pathways: linoleic acid metabolism, phenylalanine metabolism, tyrosine metabolism, nicotinate and nicotinamide metabolism and arachidonic acid metabolism. Hit metabolites in each pathway are shown in Table 2, with 15 showing upward trends and three showing downward trends (Table S1). We also found four compounds of
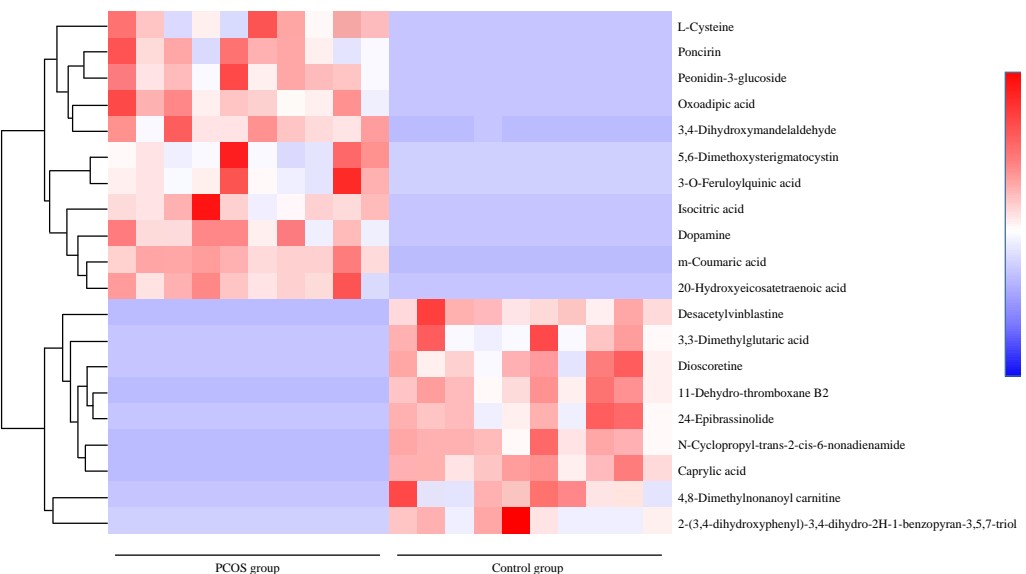

**Figure 3  Differential metabolite heat map between PCOS group and control group.** Note: Each row represented a different metabolite, each column represented for a sample. Abbreviations: PCOS, polycystic ovary syndrome.

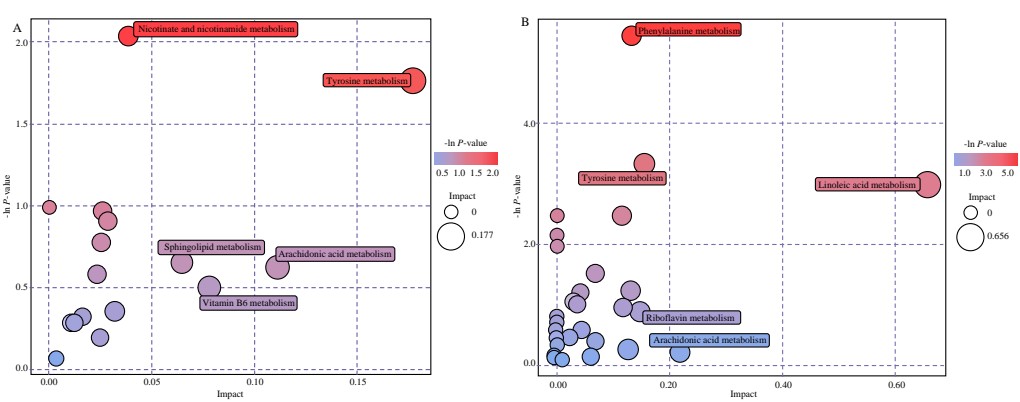

**Figure 4  Bubble diagram of metabolic pathways.** (A) Bubble diagram in positive mode. (B) Bubble diagram in negative mode. Note: Each bubble in the bubble diagram represents a metabolic pathway. Color and size of the bubble represented for -ln $P$ value and impact score (red: higher -ln $P$ value, blue: lower -ln $P$ value; large bubble: big impact, small bubble: tiny impact).

the twenty most significant metabolites in pathways. Respectively, dopamine, 11-dehydro-thromboxane B2, m-coumaric acid and 3,4-dihydroxymandelaldehyde were enriched in tyrosine metabolism, arachidonic acid metabolism and phenylalanine metabolism.

Correlation analyses were performed between differential metabolites hit in the main five pathways and sociodemographic characteristics and health indicators. As shown in Table S1 , menstruation period, dysmenorrhea, vaginal cleanness and pH had the same correlation trend and were significantly positively correlated with all hit metabolites except

**Table 2** Analysis of the top five metabolic pathways.

| Name | -Ln($P$) | Impact | Compounds | Pathways |
|---|---|---|---|---|
| **Positive ion mode** | | | | |
| Tyrosine metabolism | 1.76 | 0.18 | 3,4-Dihydroxymandelaldehyde, Homovanillin, p-Hydroxyphenylacetic acid, Homovanillic acid | hsa00350 |
| Nicotinate and nicotinamide metabolism | 2.04 | 0.04 | Niacinamide, Trigonelline, N1-Methyl-2-pyridone-5-carboxamide | hsa00760 |
| Arachidonic acid metabolism | 0.62 | 0.11 | Prostaglandin G2, 11-Dehydro-thromboxane B2 | hsa00590 |
| **Negative ion mode** | | | | |
| Linoleic acid metabolism | 2.98 | 0.66 | Linoleic acid, Bovinic acid | hsa00591 |
| Phenylalanine metabolism | 5.45 | 0.13 | Phenylacetic acid, m-Coumaric acid, Phenylpyruvic acid, 4-Hydroxybenzoic acid, p-Hydroxyphenylacetic acid | hsa00360 |
| Tyrosine metabolism | 3.31 | 0.15 | Dopamine, Gentisic acid, p-Hydroxyphenylacetic acid, Hydroxyphenyllactic acid, 3,4-Dihydroxyhydrocinnamic acid | hsa00350 |

p-hydroxyphenylacetic acid, prostaglandin G2 and 11-Dehydro-thromboxane B2. Vaginal cleanness was positively correlated with 3,4-dihydroxymandelaldehyde, N1-Methyl-2-pyridone-5-carboxamide, phenylacetic acid and 4-hydroxybenzoic acid, while educational level, BMI and fasting plasma glucose were not correlated with any hit metabolites in the top five pathways.

## DISCUSSION

As a high-resolution and high-throughput sensitive technology, non-targeted metabolomics analysis has been widely used to measure metabolite differences and explore biomarkers (*Johnson, Ivanisevic & Siuzdak, 2016*). Most of the UHPLC-MS/MS have the advantages of a comprehensive analysis range, reliable qualitative results, low detection limits and a short analysis time. We certified that PCOS group had a distinct metabolic perturbation, most of the top 20 metabolites were lipids and lipid-like molecules. All differential metabolites were targeted in five main pathways, the most important of which was linoleic acid metabolism. Many abnormal metabolic disorders were related to menstruation period (*Attia, Alharbi & Aljohani, 2023*; *Dovom et al., 2016*), while a few were related to vaginal cleanness (*Bokulich et al., 2022*). The subtle changes in vaginal microenvironment were partly explained, which may contribute to the occurrence and development of PCOS (*Bokulich et al., 2022*; *Gu et al., 2022*; *Hong et al., 2021*; *Hong et al., 2020*; *Lu et al., 2021*) and can be used as potential biomarkers for diagnosis. Meanwhile, the related metabolic pathways provided a theoretical basis for the study of the pathogenesis of PCOS in the future. It also proved that the vaginal fluid metabolomics tested by UHPLC-MS/MS showed great practicability in disease diagnosis and mechanism research.

Metabolites in human vagina are influenced by bacterial metabolism of human-derived nutrients. As samples were collected from vaginal swabs among women of reproductive age, it was considered that metabolites within vaginal microecology varied with differential metabolic activities in vivo. The results suggested that there was potential metabolic heterogeneity between PCOS patients and non-PCOS healthy controls. In this study,

differential metabolites in vaginal secretions in childbearing age women were identified, including 122 in positive ion mode and 77 in negative ion mode. Compared with previous studies, the difference in vaginal secretion samples was much greater than that in blood (*Ozegowska et al., 2021*) or follicular fluid samples (*Liu et al., 2022*). The complex environment like different compositions of vaginal and cervical mucosa microbiome may be served as a reason, which was involved with various vaginal microbiota abundance (*Wang et al., 2021*). In view of the similar metabolism of vaginal flora and human cells, it is difficult to distinguish. However, there is a lack of metabolic enzymes for the degradation of benzene ring compounds in the human body, such as oxygenase and hydrolase, and the degradation of these phenyl compounds usually depends on flora (*Somavarapu et al., 2021*). It is generally considered that the compounds containing phenyl in human metabolites are mainly produced by microbial metabolism, like 5,6-Dimethoxysterigmatocystin, m-Coumaric acid, 3,4-Dihydroxymandelaldehyde, *etc.*

Dopamine was detected as the most differential significant metabolites between PCOS and healthy controls among women of reproductive age, with a much higher level in PCOS group than in healthy controls. A study has shown a similar increased trend of dopamine in the human granulosa cells of women with PCOS (*Gómez et al., 2011*) as our study. It was previously detected in human blood and follicular fluid of ovulatory follicles of the human ovary (*Saller et al., 2014*). However, dopamine detected in vaginal swabs has been reported minimally, which revealed for the first time the presence of dopamine in human vaginal swabs. Meanwhile, dopamine is associated with cellular uptake and metabolism-dependent generation of reactive oxygen species (*Saller et al., 2014*). Additionally, dopaminergic synapse pathway is enriched by hypo-differentially methylated region genes, which are associated with PCOS (*Cao et al., 2021*). The association between dopamine and PCOS is probably led by variants in the dopamine receptor-2 gene and gonadotropin-releasing hormone dysregulation (*Chaudhari, Dawalbhakta & Nampoothiri, 2018*), which has been confirmed in animal experiments (*Amin, Horst & Gragnoli, 2023*). The first detection of dopamine in vaginal secretion reveals a method that was more convenient, economical and rapid than sampling in follicular fluid, and it is possible to screen for PCOS by detecting dopamine in vaginal secretion swabs. Also, the detection of dopamine in metabolomics may strengthen the findings of upstream omics and epigenetic factors, which would benefit future clinical targets for pharmacological study and therapeutic intervention.

Linoleic acid metabolism is the most influenced pathway by enrichment analysis of differential metabolites. Both linoleic acid and bovinic acid hit in the pathway were up-regulated in PCOS patients compared with that of healthy controls, which showed a similar trend in PCOS patients in previous plasma metabolism research (*Escobar-Morreale et al., 2012*; *Zhao et al., 2012*). Linoleic acid metabolism is associated with seborrhea, abnormal corneocyte desquamation and its storage in sebaceous follicles, suggesting that linoleic acid metabolism may be associated with PCOS obesity phenotype (*Downing et al., 1986*). In addition, linoleic acid may inhibit maturation and development of oocytes, and the enrichment of differential metabolites in linoleic acid metabolism may lead to the increase of immature oocytes and ovulatory disorder, suggesting that it may be related to the clinical symptoms of PCOS (*Marei, Wathes & Fouladi-Nashta, 2010*). Linoleic acid shows

strong pro-inflammatory activity (*Toborek et al., 2002*) and is also a potential chronic low-grade inflammatory marker of PCOS (*Escobar-Morreale, Luque-Ramírez & González, 2011*; *Ojeda-Ojeda et al., 2013*). The role of linoleic acid in the human body has already been clear, while the relationship between linoleic acid metabolism and vaginal flora still needs to be further verified in the future.

Phenylalanine metabolism plays an important role in oocyte development and ovulation (*Jóźwik et al., 2017*), and phenylalanine may be converted to tyrosine. Results showed increased tyrosine metabolism in PCOS patients with normal BMI. Tyrosine enrichment is detected in vaginal secretion in PCOS group, which shows consistency with previous studies of plasma metabolites in PCOS patients (*Zhang et al., 2014*). Additionally, a higher level of tyrosine has been previously observed in anovulatory PCOS patients in plasma metabolomics analysis (*Zhao et al., 2012*). The increase in tyrosine is related to PCOS-associated insulin resistance and ovulation dysfunction (*Fong, McDunn & Kakar, 2011*). Ovulation dysfunction is improved by lowering aromatic amino acid levels like phenylalanine and tyrosine (*Tang et al., 2019*). These results illustrate that tyrosine metabolism and phenylalanine metabolism are closely related to the pathogenesis of PCOS.

Correlations have been found between nicotinate and nicotinamide metabolism and lipid metabolism (*Yang et al., 2014*). Oral gavage for nicotinamide adapted hyperandrogenism in a rat model (*Nejabati et al., 2020*). However, it has been proved that there remains an optimum dose of nicotinamide, or that it would be harmful to homeostasis of glucose and damage insulin resistance (*Cantó et al., 2012*). Nicotinamide acts as the substrate of N-methyltransferase, which leads to the generation of N1-Methylnicotinamide. Hence the further formation of N1-Methyl-2-pyridone-5-carboxamide. Studies have shown that the production of N1-Methylnicotinamide in cumulus cells of patients with PCOS was significantly increased (*Nejabati et al., 2020*). The upstream and downstream products of N1-Methylnicotinamide are tested to be increased, which indicates that nicotinate and nicotinamide metabolism can also be detected in vaginal secretions, as in tissues and serum.

Linoleic acid can be converted into arachidonic acid by acyl-CoA 6-desaturase, elongation of very long chain fatty acids protein 5 and acyl-CoA (8-3)-desaturase. Arachidonic acid is then converted to a series of short-lived metabolites. In arachidonic acid metabolism, arachidonic acid and its cyclooxygenase-generated metabolites have the ability to regulate different ovarian functions and luteolysis (*Husein & Kridli, 2003*; *Medan et al., 2003*). Studies have shown that the increase of arachidonic acid and linoleic acid in follicular fluid significantly decreased the ability of oocytes to form nucleus and fertilize, which may be related to the low fertility of patients with PCOS. However, the level of arachidonic acid changes in different tissues. In the PCOS rat model, arachidonic acid in serum was up-regulated while down-regulated in ovarian tissue (*Huang et al., 2018*). Arachidonic acid derived metabolites, such as prostaglandins and thromboxane, have been shown to play an important role in oocyte maturation, cumulus expansion and ovulatory (*Li et al, 2019*). Meanwhile, linoleic acid, arachidonic acid and their downstream metabolites are also associated with cardiovascular diseases (*Sonnweber et al., 2018*), carcinogenesis (*Sah et al., 2022*), inflammatory diseases (*Li et al., 2020*), which have been considered the long-term

complications of PCOS. For example, arachidonic acid metabolites would contribute to sclerotic vessels (*Sonnweber et al., 2018*) and inflammation in multiple organs (*Shen et al., 2019*). Focusing on PCOS and its chronic and complex complications may help to decrease the impairment in organ functioning. In this study, 11-Dehydro-thromboxane B2 and prostaglandin G2, which were downstream products of thromboxane B2 and upstream products of prostaglandin A2, prostaglandin B2, prostaglandin C2 and prostaglandin E2, respectively, further indicated the important role of arachidonic acid metabolism in PCOS patients.

**Limitations**

First, we used stricter inclusion criteria for both PCOS and healthy controls, which means PCOS women should be newly diagnosed and meet both Rotterdam criteria and Chinese Guidelines for Diagnosis of PCOS to ensure typical PCOS cases and healthy controls were followed up for three months for regular menstruation and to exclude complications, which limited the study sample size. Under this circumstance, the small sample size limited the statistical efficiency due to the pilot investigation design. Type II error may be possible and further generalization should be done with caution. Second, phenotypic of PCOS was not included in the pilot study. In this case, a wider range of different metabolomics profiles was detected. Third, although age match and ethnicity match were conducted through our study to eliminate more baseline covariate differences among groups, potential covariates between groups like nutrient requirement and food intake habits were still inevitable. Fourth, as PCOS women had irregular menstrual cycles, time in swab collection was hard to classify further into follicular, ovulation and luteal phases. There might be differences in metabolites in varied phases of menstrual cycles. Fifth, since merely Chinese Han reproductive age women were included in a single center study, extrapolation to other ethnicities should take extreme caution.

# CONCLUSIONS

Based on this reproductive age pilot investigation, we found significant metabolomics differences between PCOS and healthy controls. Vaginal metabolites, especially dopamine, can be regarded as a potential biomarker in PCOS screening, and linoleic acid metabolism can be identified as the most influenced pathway. This study highlights the need for vaginal secretion metabolism in reproductive age and for careful personalized health diagnosis and potential targets for therapeutic intervention.

**List of abbreviations**

| | |
|---|---|
| **ESI** | electrospray ionization |
| **OPLS-DA** | orthogonal partial least-squares discriminant analysis |
| **PCOS** | polycystic ovary syndrome |
| **UHPLC-MS/MS** | ultra-high-performance liquid chromatography tandem mass spectrometry |

# ACKNOWLEDGEMENTS

We thank all the countless participants and health workers for their tremendous efforts and collaboration.

### Funding

This research has received funding from the National Natural Science Foundation of China (Grant No. 81872634), People's Republic of China. The funders had no role in study design, data collection and analysis, decision to publish, or preparation of the manuscript.

### Grant Disclosures

The following grant information was disclosed by the authors:
National Natural Science Foundation of China: 81872634.

### Competing Interests

The authors declare there are no competing interests.

### Author Contributions

- Yan Xuan conceived and designed the experiments, performed the experiments, analyzed the data, prepared figures and/or tables, and approved the final draft.
- Xiang Hong conceived and designed the experiments, performed the experiments, analyzed the data, authored or reviewed drafts of the article, and approved the final draft.
- Xu Zhou performed the experiments, analyzed the data, prepared figures and/or tables, and approved the final draft.
- Tao Yan performed the experiments, analyzed the data, prepared figures and/or tables, and approved the final draft.
- Pengfei Qin performed the experiments, prepared figures and/or tables, and approved the final draft.
- Danhong Peng performed the experiments, prepared figures and/or tables, and approved the final draft.
- Bei Wang conceived and designed the experiments, authored or reviewed drafts of the article, and approved the final draft.

### Human Ethics

The following information was supplied relating to ethical approvals (i.e., approving body and any reference numbers):

This study has been approved by the Ethics Committee of Zhongda Hospital (2018ZDSYLL072-P01).

### Data Availability

The raw data is available in the Supplemental Files and at figshare: Xuan, Yan (2024). raw data.xlsx. figshare. Dataset. https://doi.org/10.6084/m9.figshare.26539447.v1.

## Supplemental Information

Supplemental information for this article can be found online at http://dx.doi.org/10.7717/peerj.18194#supplemental-information.

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
