# Peer review of "The vaginal metabolomics profile with features of polycystic ovary syndrome: a pilot investigation in China"

_PeerJ, doi:10.7717/peerj.18194_

## Round 0.1 · original submission · Major Revisions

Thank you for the submission which has been looked at by independent reviewers. Both of whom are positive about this work. However, one in particular raises some issues which must be addressed and will require a substantial re-write of the paper to reflect issues around sample size.

Your study involves only ten women with PCOS and ten matched healthy controls. This small sample size severely limits the statistical power of the metabolomic analysis. With such a limited number of participants, detecting subtle but potentially important differences in metabolite levels becomes challenging, and the results may not be generalizable to the broader population. Small sample sizes increase the risk of Type II errors (stats) and reduce the reliability and reproducibility of the findings. Both reviewer-1 and I advocate strongly that you clearly and unambiguously acknowledge this limitation and reframe the study as a pilot investigation, as early in the paper as the abstract, and clearly then throughout the intro and discussion. We do not believe this will weaken the impact of your study, but think it is important that this is acknowledged ‘up-front’.

The introduction lacks depth and a critical evaluation of current PCOS research. The problem identification is not sufficiently supported by statistics, including both clinical and societal impacts of PCOS. The authors often assume that readers have a theoretical understanding of PCOS and the technical aspects of their analysis, which can be frustrating. The metabolomics section is well-received, but it is important to remind the authors that metabolomics is merely a stepping stone. The role of epigenetics, especially in the development of PCOS, should not be overlooked. Thus, the paper lacks contextualization of environmental influences and the underlying genetic contributions to PCOS. The references cited are often outdated, with only one article from 2023, highlighting the need for more current literature to support key statements.

Both reviewers have made specific suggestions in their reports, and one has taken the time to provide an annotated paper which you should read carefully and fully address all points noted.
Please be clear that we do not require further experimental work and recognise the interest this study will generate. The suggestions we make are driven by a need for clarity and a desire to see this work framed in its best possible light. Hence, although this is a ‘major revision’ reflecting the extent of the changes, we think this will require only time and effort with a keyboard and not further experimental analysis!

·

Basic reporting

I appreciate the effort put into your study titled "The vaginal metabolomics profile with features of polycystic ovary syndrome: a matched case-control study in China." This research addresses an important area and has the potential to contribute significantly to our understanding of PCOS. However, there are several areas where the article falls short of the standards expected by PeerJ.
Firstly, the introduction lacks depth and fails to sufficiently contextualize the significance of PCOS. It would benefit from more case studies and a critical evaluation of current research. Additionally, the problem identification needs stronger statistical support, including the clinical and societal impact of PCOS. The rationale for choosing vaginal secretions for analysis should be better contextualized, and more details about the affected age group and geographic distribution are needed.

Experimental design

The dual aim of the study appears overly ambitious and broad. Narrowing the focus to specific outcomes would make the study more targeted and manageable. The limitations section does not adequately address the significant impact of the small sample size on the study's statistical power. Including only ten women with PCOS and ten healthy controls limits the ability to detect important differences in metabolite levels and reduces the generalizability of the results. This study should be framed as a pilot investigation to reflect these limitations accurately.

Validity of the findings

The metabolomics section is well-described, but it lacks substantial contextualization of DA within vaginal omics and its clinical relevance. Moreover, the manuscript fails to correlate findings with genetic and epigenetic factors, which are crucial given the interrelation between metabolites and genetic influences. Furthermore, while the discussion section is well-written, the manuscript would benefit from a more thorough explanation of the clinical significance of the findings and their potential use in pharmacological studies. There is a need to build upon the phenotypic considerations introduced and to integrate recent studies to validate the findings.

Additional comments

Overall, your study is impactful and addresses a critical area in PCOS research. Ensuring that the study is framed appropriately and providing additional context and detail will strengthen this article. Additionally, incorporating more recent references and addressing the limitations of your study transparently is needed.

Reviewer 2 ·

Basic reporting

Clear and Concise. The manuscript is written in clear and understandable language.

The title of manuscript states with features of PCOS, the features and presentation of PCOS varies, for example in Rotterdam criteria, the presence of two of three of the criteria such as oligo‐anovulation, hyperandrogenism and polycystic ovaries must be present. Does all women in this study presents with same features or different?

Row161, 162, Please clarify the sentence. In PCOS cycles are usually irregular, the proportion is donated to all participants or only PCOS? not clear

In row no 181, 182, DA was the top significant metabolite in vaginal secretions of which group? It should be mentioned.

In row 212, please provide a valid reference on this statement: Many abnormal metabolic disorders were related to menstruation period, while a few were related to vaginal cleanness. Are these authors findings?

What is the evidence that vaginal microenvironment may contribute to the occurrence and development of PCOS. Is this statement by authors findings or it has been stated in the Literature, if so, kindly provide a valid reference.

Row 218, authors have highlighted that vaginal microecology vary with differential metabolic activities in vivo, wonder if there were any difference in the nutrient requirement and food intake habits of study participants. please provide a valid ref for this statement.
Row 227 to 231, again provide ref for this statement.
The authors have clearly mentioned that nutrient requirements, metabolism and vaginal microflora may also varies in between women. Does this mean that metabolic pathways and vaginal microenvironment in PCOS and non PCOS also varies?

Row 230, 240. The author highlighted that DA can be used as a potential biomarker to diagnose PCOS, can it also be used as a prognostic marker? how it can be beneficial from a therapeutic point of view. if authors can elaborate a bit more.

Row 243, Plasma of whom, which population? Is it in PCOS (which was similar to previous research in plasma metabolism)?

Row 255 and 256, 257Tyrosine enrichment was detected in vaginal secretion of which group? PCOS?

Experimental design

Experimental design is good. to find specific age matched control is unlikely and therefore the number of participants is 20.

The only limitation is that authors didn't highlight about which day of menstrual cycle samples were collected? It is likely that some women might be in proliferative stage and some in ovulation, and/or secretory phases. There could be a huge difference in metabolites in different phases of menstrual cycle.

It is difficult to interpret the phase of menstrual cycle in irregular cycles of PCOS, however when researchers were collecting samples, they could have analyzed vaginal secretions under microscope which could have been of some help.

Please separate Inclusion and Exclusion criteria in a separate heading for better read.

Validity of the findings

The authors have looked into metabolomics in vaginal fluid, but why it is significant? Presence of these nutrients' metabolites can help in what way? This can be elaborated further.


For example, linoleic acid is converted to gamma-linolenic acid and further to arachidonic acid. AA is a precursor in the formation of leukotrienes, prostaglandins, and thromboxane. Their role in reproductive system is elaborated by the authors in the discussion section, however, could this also be associated with cardiovascular diseases, carcinogenesis, inflammatory diseases etc in women with PCOS?
As PCOS women usually seeks medical based on menstrual irregularity and infertility issues which are resolved with medications, but the long-term complications of PCOS which may affect different organ function and their correlation with these metabolites can be highlighted. it is a suggestion.

Authors have highlighted the limitations at the end of paper, which is good.

---

## Round 0.2 · Minor Revisions

Below is the response from one of the reviewers:

I appreciate the authors' diligent efforts in addressing the feedback I previously provided on their manuscript titled "The vaginal metabolomics profile with features of polycystic ovary syndrome: a matched case-control study in China." The authors have effectively revised the manuscript to present the study as a pilot investigation, which I believe is appropriate given the limited sample size. This change enhances the clarity and contextualization of the study's scope and limitations, addressing my primary concern regarding the statistical power and generalizability of the findings.

The introduction has been significantly improved, with a deeper exploration of current PCOS research and the inclusion of relevant statistics to support problem identification.

Regarding epigenetics, the authors have made an attempt to address my concerns. However, I acknowledge that the depth of their discussion remains somewhat limited, likely due to constraints in their research expertise. It is clear that the authors are more comfortable with the metabolomics aspect of the study, which they have handled well. While the exploration of genetic factors is still not as robust as I would prefer, the revisions made are satisfactory given the focus of their research.

I noticed instances of informal writing in the manuscript, such as the use of phrases like "and so on." I recommend that these be revised to more precise and formal expressions to maintain the scholarly tone. The discussion and results sections have also seen improvements, aligning better with the narrative of a pilot study.

The manuscript now feels more cohesive, and the additional references have strengthened the overall quality of the work. In summary, I believe the manuscript has been substantially improved and is now suitable for publication, pending any final editorial considerations.

Given this, could you make a final pass through your study and address issues around informal English, then re-submit? This should not take you long and I will then accept the paper.

---

## Round 0.3 · accepted · Accept

Thank you for attending to these edits. I am delighted to accept this now.